# Quantitative Measurement of Breast Tumors Using Intravoxel Incoherent Motion (IVIM) MR Images

**DOI:** 10.3390/jpm11070656

**Published:** 2021-07-13

**Authors:** Si-Wa Chan, Wei-Hsuan Hu, Yen-Chieh Ouyang, Hsien-Chi Su, Chin-Yao Lin, Yung-Chieh Chang, Chia-Chun Hsu, Kuan-Wen Chen, Chia-Chen Liu, Sou-Hsin Chien

**Affiliations:** 1Department of Medical Imaging, Taichung Tzu Chi Hospital, Buddhist Tzu Chi Medical Foundation, Taichung 427, Taiwan; tc1683101@tzuchi.com.tw (S.-W.C.); jiajium@tzuchi.com.tw (C.-C.H.); flydragonk@tzuchi.com.tw (C.-C.L.); 2School of Medicine, Tzu Chi University, Hualien 970, Taiwan; gbjeff1014@tzuchi.com.tw (C.-Y.L.); tc1691401@tzuchi.com.tw (K.-W.C.); shchien@tzuchi.com.tw (S.-H.C.); 3Department of Medical Imaging Radiological Science, Central Taiwan University of Science and Technology, Taichung 427, Taiwan; 4Department of Electrical Engineering, National Chung Hsing University, Taichung 402, Taiwan; tommy357099@smail.nchu.edu.tw (W.-H.H.); g108093005@mail.nchu.edu.tw (H.-C.S.); 5Department of Surgery, Taichung Tzu Chi Hospital, Buddhist Tzu Chi Medical Foundation & School of Medicine, Taichung 407, Taiwan; 6Department of Radiology, Taichung Veterans General Hospital, Taichung 407, Taiwan; vincent760115@vghtc.gov.tw; 7Department of Radiation Oncology, Taichung Tzu Chi Hospital, Buddhist Tzu Chi Medical Foundation, Taichung 427, Taiwan; 8Division of Plastic Surgery, Taichung Tzu Chi Hospital, Buddhist Tzu Chi Medical Foundation, Taichung 427, Taiwan

**Keywords:** magnetic resonance imaging, intra-voxel incoherent motion, hyperspectral image cube, iterative-constrained energy minimization, kernel-constrained energy minimization, K-means, fuzzy C-means

## Abstract

Breast magnetic resonance imaging (MRI) is currently a widely used clinical examination tool. Recently, MR diffusion-related technologies, such as intravoxel incoherent motion diffusion weighted imaging (IVIM-DWI), have been extensively studied by breast cancer researchers and gradually adopted in clinical practice. In this study, we explored automatic tumor detection by IVIM-DWI. We considered the acquired IVIM-DWI data as a hyperspectral image cube and used a well-known hyperspectral subpixel target detection technique: constrained energy minimization (CEM). Two extended CEM methods—kernel CEM (K-CEM) and iterative CEM (I-CEM)—were employed to detect breast tumors. The K-means and fuzzy C-means clustering algorithms were also evaluated. The quantitative measurement results were compared to dynamic contrast-enhanced T1-MR imaging as ground truth. All four methods were successful in detecting tumors for all the patients studied. The clustering methods were found to be faster, but the CEM methods demonstrated better performance according to both the Dice and Jaccard metrics. These unsupervised tumor detection methods have the advantage of potentially eliminating operator variability. The quantitative results can be measured by using ADC, signal attenuation slope, D*, D, and PF parameters to classify tumors of mass, non-mass, cyst, and fibroadenoma types.

## 1. Introduction

According to statistics recently published by the Ministry of Health and Welfare, the incidence of breast cancer continues to rise in Taiwan; it has gradually become the most important health issue for women. Several studies in Europe and the United States show that breast cancer can be detected by regular screening [1]. Early diagnosis and treatment of breast cancer can significantly reduce the mortality rate. Breast magnetic resonance imaging (MRI) is a widely used screening tool today. Several studies have shown that the sensitivity of MRI can be higher than 90%, significantly better than the sensitivity of mammography and ultrasonography [2,3].

A typical MRI scanner consists of a main magnetic field, a gradient magnetic field, a radio frequency system, and a computer system. Strong magnetic fields and radio frequency (RF) pulses are used to convert the distribution of hydrogen atoms into information about the anatomical structure of the human body on computer-generated images. The human body is rich in water molecules, so the hydrogen atoms in the water molecules can be used as a source of signals. Since the resonance effect of protons in different types of tissues will produce different signal intensities, we can determine the location of each tissue. Generally speaking, the stronger the magnetic field, the higher the resolution of MRI. Unlike other screenings, MRI can apply different pulse sequences to generate images with different weights. Comparing differently weighted images and multiplanar scan views of the cross-section of the lesion is very useful for diagnosis. Breast MRI has a high cancer detection rate [4]. For some locally invasive breast cancers, MRI also has a good performance in assessing the effect of preoperative chemotherapy [5].

Diffusion-weighted imaging (DWI) is a commonly used technique in MRI screening [6,7]. By measuring the displacement of water molecules due to Brownian motion, DWI explores the tissue microstructure. The low diffusion coefficient of highly cellular tissues renders this technique particularly useful in tumor characterization. Recently, intravoxel incoherent motion (IVIM) diffusion weight imaging (DWI) has been developed. The bi-exponential model of IVIM-DWI can obtain the pure diffusion coefficient, D, of water molecules, the pseudo-diffusion coefficient, D*, and the perfusion fraction, PF. In addition, the model can distinguish between water molecular diffusion and microcirculation perfusion in tissue. Current techniques for measuring perfusion in tissue, such as dynamic contrast-enhanced MRI (DCE-MRI), are generally dependent on the use of exogenous contrast agents. Hemodialysis patients may experience a decline in renal function after the use of contrast agents, which may further lead to renal fibrosis. In contrast to the invasive use of contrast agents, IVIM-DWI is a non-invasive technique that uses magnetic resonance imaging to observe the diffusion of water molecules in living tissue. It has been reported that IVIM-DWI parameters can identify both benign and malignant breast lesions [8,9,10], showing that IVIM-DWI has advantages over traditional DWI [11,12]. In our previous study, IVIM-DWI was used in combination with hyperspectral techniques and the IVIM-DWI MR data were considered as a hyperspectral image cube [12]. We here consider IVIM-DWI with 13 different weighting factors (*b*-values) as a 13-band multispectral image cube in which each sequence-acquired image is viewed as a spectral band image. Kernel-constrained energy minimization (K-CEM), iterative-constrained energy minimization (I-CEM), K-means, and fuzzy C-means are all used in this study for tumor detection. K-means and fuzzy C-means are two clustering methods used to separate signals (in our case, images) into several classes. Thus, in this study, we compare two approaches (energy minimization and clustering) and two ways of implementing each approach. Further, we explore the efficiency of the four different tumor detection methods and their clinical applicability.

This produced IVIM-DWI data as a hyperspectral image cube and used a well-known hyperspectral subpixel target detection technique: constrained energy minimization (CEM). Two extended CEM methods—kernel CEM (K-CEM) and iterative CEM (I-CEM)—were employed to detect breast tumors. The K-means and fuzzy C-means clustering algorithms were also tested. The results were then compared to dynamic contrast-enhanced T1-MR imaging as ground truth. All four methods were successful in detecting tumors for most patients studied.

## 2. Methodology and Methods

### 2.1. Diffusion-Weighted Imaging (DWI)

DWI is a form of MR imaging based on the measurement of the random Brownian motion of water molecules in body tissue. The concept of DWI was developed by Stejskal and Tanner [13]. Water is the main constituent of the human body. The cell walls in the human body form natural barriers blocking the free movement of water molecules. The motion of intracellular water molecules is therefore restricted in comparison with the motion of extracellular water molecules; this is especially true in areas with dense cells. Therefore, by measuring the motion of water molecules in the body, we can analyze the cell structures within a tissue. A commonly used scanning sequence involves applying a pair of bipolar-pulsed gradients. Subsequently, the signal intensity from the water molecules begins to decay. The rate of decay reflects the rate at which the water molecules diffuse; therefore, the signal intensity from regions with many obstacles is stronger on the diffusion-weighted image than that from regions where water molecules move freely [14].

### 2.2. Intravoxel Incoherent Motion (IVIM) Imaging

IVIM was proposed by Le Bihan et al. [15], who believed that motion in human tissue would cause phase changes within a voxel and produce echo attenuation in MRI. In this context, the term “motion” includes both the real diffusion of water molecules and microcirculation perfusion within tissue. The capillary network is pseudo-randomly distributed in the voxel; hence, the diffusion of water molecules in the blood vessel is similar to the diffusion of free water molecules in a single voxel. The probability of diffusion is equal in all orientations; therefore, microcirculation perfusion may be considered as pseudo-diffusion. Although the fluid flowing in the capillary is only a small fraction of the total water content in normal tissues (such as brain tissue), the pseudo-diffusion coefficient of the microcirculation perfusion in the capillary is several times larger than the pure diffusion coefficient of the water molecules. For theoretical analysis, Le Bihan et al. proposed the IVIM-DWI model (bi-exponential model) and a corresponding imaging method [16]. In this model, the diffusion of water molecules and microcirculation perfusion are distinguished. The bi-exponential model of IVIM is defined using the expression:(1)$$\frac{S_{b}}{S_{0}}=(1-PF)\exp(-b\cdot D)+PF\exp(-b\cdot D^{*})$$
where *b* is the diffusion sensitization of the MRI sequence; *S**_b_* and *S*_0_ are signal intensities in pixels with and without the diffusion gradient, respectively; PF is the perfusion fraction, i.e., the volume ratio in the voxel between the microcirculation perfusion effect and the total diffusion effect; D is the normal diffusion coefficient for simple diffusion of water molecules in a voxel; and D* is a pseudo-diffusion coefficient corresponding to the microcirculation perfusion-related diffusion motion in a voxel. The ADC can be calculated if PF = 0 then D = ADC.

The ADC value was measured by at least two DW imaging (DWI) with different signal attenuation at different *b* values. The values displayed reflect the degree of diffusion of water molecules through different tissues, according to the following equation:(2)$$\ln(S_{b}/S_{0})=-b\times (ADC)\;\to \;ADC=-1/b\times \exp(S_{b}/S_{0})$$

### 2.3. Band Expansion Process (BEP)

BEP, proposed by Ren and Chang [17], was originally used in the field of hyperspectral remote sensing. If we consider the original MR images as random variables, we can create a set of second-order statistical band images by finding the correlation between different MR band images [18]. With these correlated images, we can use autocorrelation, cross-correlation, and nonlinear correlation to obtain useful second-order statistics that were previously missing. Therefore, nonlinear correlation images produced by the BEP can assist in increasing the accuracy of detection [12].

### 2.4. Automatic Target-Generation Process (ATGP)

The automatic target generation process (ATGP) [19] is an unsupervised process that generates a set of targets from image data, which are subsequently classified by the target classification process (TCP) [13]. For fully automatic detection, the ATGP method was used to find the initial sample of suspicious lesions automatically on the IVIM data with the hyperspectral imaging method, and the breast tissue was classified. The vectors generated pixel by pixel in the hyperspectral were used as training data, which could make the detection of the target edge more accurate.

### 2.5. Spectral Angle Mapper (SAM)

SAM is a method based on spectral classification for directly comparing image spectra and target spectra [20,21,22]. This method determines the extent of similarity between two different spectra (represented as vectors) by calculating the angle between them. The smaller the angle between the two spectra, the higher the similarity [13].

### 2.6. Constrained Energy Minimization (CEM)

CEM has been widely used for target detection in hyperspectral remote sensing imaging [12]. It detects the desired target-signal source by using a unity constraint while suppressing noise and unknown signal sources and minimizing the average output energy [23]. CEM [24] was developed for the cases in which the only required knowledge is the signature to be classified.

### 2.7. K-means Clustering and Fuzzy C-Means (FCM) Clustering

K-means is a well-known unsupervised learning algorithm for solving the clustering problem [25,26]. Because it is simple, efficient, and easy to implement, K-means has been widely used.

The FCM clustering algorithm was proposed by Dunn in 1973 and later improved by Bezdek [27,28,29].

### 2.8. Kernel-Based Constrained Energy Minimization Approach (K-CEM)

Constrained energy minimization is a linear filter and IVIM images are nonlinearly generated multispectral images; therefore, nonlinear separability may present an issue for CEM. To mitigate this problem, CEM was further expanded to a kernel version of CEM, called kernel CEM (KCEM) [30], which makes the target detection task more effective [12].

### 2.9. Evaluation of the Detection Method

To evaluate the detection performance, we used the standard *Dice* similarity coefficient, representing the degree of overlap between two binary images [31], and the *Jaccard* similarity coefficient, an indicator of the similarity between the two sets [32]. These can be expressed as
(3)$$Dice(A,B)=\frac{2\left|A\cap B\right|}{\left|A\right|+\left|B\right|}$$
and
(4)$$Jaccard(A,B)=\frac{\left|A\cap B\right|}{\left|A\cup B\right|}$$
where *A* is the automatically segmented image and *B* is the ground truth image. The ground truth tumor images were segmented by three experienced radiologists.

### 2.10. Confusion Matrix

We also generated the confusion matrix to evaluate the detection. Positive samples that are correctly predicted as positive are denoted as *TP*. Negative samples that are correctly predicted as negative are denoted as *TN*. When negative samples are wrongly predicted as positive, they are denoted as *FP*. When positive samples are wrongly predicted as negative, they are denoted as *FN*. We can then evaluate the precision rate, accuracy rate, and recall rate. The precision rate is defined as all the positive prediction samples that are positive samples:(5)$$Precision=\frac{TP}{TP+FP}$$

The accuracy rate is defined as the prediction samples that can be correctly predicted:(6)$$Accuracy=\frac{TP+TN}{TP+TN+FP+FN}$$

The recall rate is defined as all the positive samples that can be predicted:(7)$$Recall=\frac{TP}{TP+FN}$$

## 3. Experimental Process

### 3.1. Experimental Materials

MR imaging was performed on a 3T system (Discovery MR750 3.0 Tesla, GE, Boston, MA, USA) with an 8-channel HD breast coil. Axial IVIM images with bilateral breast coverage were acquired (repetition time (TR) = 3000 ms, echo time (TE) = 74 ms, field of view (FOV) = 320 mm, flip angle (FA) = 90°, inversion time (TI) = 115 ms, matrix = 108 × 108, and reconstructed voxel size = 3 × 3 × 4 mm^3^). The axial T1-weighted DCE-MRI (TR = 1.0 ms, TE = 4.3 ms, FOV = 320 mm, FA = 10°, TI = 99 ms, voxel = 320 × 320 × 20, and slice thickness = 2 mm) was acquired using the gradient echo sequence technique. DW imaging was conducted with 13 *b* values (0, 15, 30, 45, 60, 100, 200, 400, 600, 1000, 1500, 2000, and 2500 s/mm^2^). Figure 1 shows IVIM-DW MR images for all 13 *b* values. Forty-eight cases with breast tumors underwent MR examinations at Taichung Tzu Chi Hospital, Taichung, Taiwan, including 29 cases with mass, 17 with non-mass, 1 with cysts, and 1 with fibroadenoma. The tumor location, size range, tumor type, etc. of all IVIV-MR medical images were manually marked by experienced radiologists and surgeons from dynamic contrast-enhanced T1-MR. The actual diagnosis of the tumor was also checked by ultrasound images and confirmed by biopsy. The IVIM-MR image data were considered to form a 3D image cube where each image pixel is a three-dimensional vector (voxel), of which each component is a pixel specified by one particular image sequence.

### 3.2. Image Pre-Processing

The steps of pre-processing are shown in Figure 2; further details are explained in this section. First, we used the N3 technique to adjust the non-uniformity of the MR images. This method is independent of pulse sequence and is insensitive to pathological data that might otherwise violate model assumptions. Pre-processing of MR data using N3 has been shown to substantially improve the accuracy of anatomical analysis techniques such as tissue classification and cortical surface extraction.

Because of the displacement of multiple *b*-value images of IVIM owing to fluctuations from breathing, we made an adjustment match for all the pixels using the Avizo software package [30]. Finally, the definition of the range of the breast was analyzed and undesirable skin and muscles were removed from the images.

### 3.3. Tumor Detection Using CEM Methods

In this section, we explain the use of the K-CEM and I-CEM clustering algorithms to detect breast tumors from hyperspectral images.

The CEM algorithm is a supervised method. Therefore, an algorithm providing spectral features of the target of interest is needed when using the CEM algorithm to detect breast tumors. First, the spectral signature of the object of interest (in this study, a tumor) is found. The signal intensity of tumors in diffusion-weighted imaging is significantly different (usually higher) from that of normal breast tissue. Therefore, for our high-intensity spectral signature of interest, we used the ATGP algorithm to find high-intensity initial training samples. The result of the CEM algorithm depends on the selection of training samples because the CEM algorithm is sensitive to the target of detection. To prevent extreme results from an unrepresentative initial selection, we used SAM to find other samples similar to the initial training samples. Subsequently, all these samples were averaged to generate a final training sample for CEM.

Figure 3 is a flowchart depicting the steps involved in tumor detection via I-CEM. I-CEM avoids unstable CEM results owing to the choice of the initial training sample. First, we executed CEM with the initial training samples and obtained a preliminary classification result. Next, ten percentage points of the average from the first classification result were randomly selected as new training samples. The result was compared with the previous result to calculate the difference rate. If the difference rate exceeded 1%, we repeated the same steps; otherwise, we stopped the iterative process and recorded the result.

Figure 4 is a flowchart depicting the steps involved in tumor detection via K-CEM. K-CEM employs various kernels to expand the original data space to a higher dimensional feature space where CEM is then used. In addition, the kernel that was used in this experiment was a Gaussian radial basis function (RBF) kernel. Then, Otsu’s method of thresholding is used to create a binary image of the tumor.

The original IVIM dataset contains 13 *b* value slices, which can be considered as a set of 13-bandmulti-spectrum images. We used BEP (see Figure 5) to generate more bands on IVIM-DW images so that the multi-spectrum method was more efficient. Using cross-correlation and autocorrelation, the number of bands can be expanded from 13 to 104 bands, which provide useful second-order statistical information. The 104 bands are generated from 13 original bands, 13 auto-correlation bands, and 78 cross-correlation bands.

### 3.4. Tumor Detection Using Clustering Methods

In this section, we explain how to use the K-means and fuzzy C-means algorithms to detect breast tumors. Unlike traditional MR breast image segmentation [7], the original IVIM data set contains 13 *b*-value slices. Before processing the data, a technique called non-parametric non-uniform intensity normalization (N3) is used to adjust the non-uniformity of the MR image. Multiple b-value images of IVIM displacement caused by breathing must be corrected; therefore, alignment is used to register all pixels. After aligning all the images, a method to automatically extract the breast area was developed [12].

As shown in Figure 6 and Figure 7, an automatic target generation process (ATGP) is used to identify the cluster containing the tumor regions.

K-means clustering is one of the simplest unsupervised learning algorithms and is widely applied to clustering analysis. It provides a simple and easy way to classify n given data points into k subsets by minimizing an objective function, and can be applied to tumor segmentation. The K-means clustering algorithm classifies each pixel of an IVIM-MR image into K clusters, where the number K is selected manually. In Figure 8, we set the *k* number of clusters to a range of 2–6. Then, ATGP is used to identify the cluster containing the tumor region; the processing flowchart is shown in Figure 6. We generated a new figure to show the results of clustering using K-means from when *k* = 2 to *k* = 6. Figure 8a shows clustering results with the band expansion process. Figure 8b shows the clustering results without the band expansion process. From these cluster images, we observe that *k* = 3 depicts the best cluster image. We observe in Figure 8 that the breast lesion of the IVIM-MR image is expected to be classified when the *k* number is set to at least three parts, including normal tissue, tumor, black background, and possibly any other parts. The image area that was pointed out by ATGP usually has abnormal tissue. Similar steps were followed by using the FCM clustering algorithm to classify each pixel of an IVIM-MR image into three clusters. Moreover, each pixel of the IVIM-MR image is associated with each cluster by a fuzzy membership function. Similarly, we used ATGP to identify clusters containing tumor areas, see Figure 7. Finally, a grayscale image of tumors is converted into a binary image by Otsu’s method.

## 4. Experimental Results

In this study, we used the aforementioned four automatic detection methods for breast tumor detection. Dynamic contrast-enhanced T1 imaging plays an important role in diagnosis of breast tumors and was therefore used as a reference for the comparison of the results. Next, the Dice and Jaccard similarity coefficients and the execution time for each method was used to evaluate the efficiencies of the four detection methods. Further, we conducted experiments using real images to establish the utility of the methods for real clinical cases.

### 4.1. Real Case of Mass Tumor

This case is of a patient who was diagnosed with a mass-type tumor in her right breast. We used K-CEM, I-CEM, K-means, and FCM for the tumor detection. Furthermore, these results were compared with dynamic contrast-enhanced T1-MR imaging. The variation in the decay of the signal intensity of the tumor was detected by the four methods using different *b* values. Figure 9 shows third-slice IVIM-DW images of a mass tumor and the results of tumor detection via the four methods, as well as the mapping of the tumor region by dynamic contrast-enhanced T1-MR imaging. The clustering methods are faster, but the CEM methods have better performance on other metrics. The two clustering methods (K-means and FCM) were highly efficient and show the potential for tumor detection with a short execution time. These unsupervised methods are beneficial for tumor detection, as operator variability is potentially eliminated.

Figure 10 shows the attenuation of the tumor signal intensity detected by these four methods using different *b* values in the third slice. From this figure, we can see that all four signal strength curves are very similar and almost in the same curve. This means that the same conclusion can be drawn for the tumors detected by the four methods using different b values. The red area in Figure 10 was enlarged on the top of Figure 10.

Table 1 shows ADC, signal decay slope, D*, D, and PF parameter values for a mass tumor obtained by using different detection methods: K-CEM, fuzzy C-means, K-means, and I-KCEM.

### 4.2. Real Case of a Non-Mass Tumor

This case is of a patient who was diagnosed with a non-mass type tumor in her right breast. Figure 11 shows the 8th-slice IVIM-DW images of the non-mass tumor and the results of tumor detection by the aforementioned four methods, as well as the mapping of the tumor by dynamic contrast-enhanced T1-MR imaging. Figure 12 displays the signal-intensity decay of the tumor detected by the four methods with different *b* values in the 8th slice.

Table 2 shows ADC, signal decay slope, D*, D, and PF parameter values for a non-mass tumor obtained by using different detection methods: K-CEM, fuzzy C-means, K-means, and I-KCEM.

### 4.3. Real Case of a Fibroadenoma

Breast adenofibroma, also known as breast fibroadenoma, is the most common benign tumor found in a woman’s breast. It is prevalent among women in the 15–35 age group, especially those around the age of 20. It is, in general, an isolated solitary mass in the breast, usually between 1 and 3 cm in size, although, in approximately 20% of cases, it consists of multiple masses appearing simultaneously or consecutively in both breasts. The mass, with a smooth surface and a tough and elastic texture, is well-defined and easily pushed into the breast. Although breast fibroadenoma is benign in most cases, it can sometimes be malignant. Therefore, once a breast fibroid is found, it should be removed by surgery. In some cases, fibroadenoma will recur at the original surgical site or regenerate as new adenomas elsewhere in the breast.

We consider a patient who was diagnosed with breast fibroadenoma in her right breast. Figure 13 shows the 8th-slice IVIM-DW images of the fibroadenoma and the results of fibroadenoma detection by the aforementioned four methods, as well as the mapping of the fibroadenoma region by dynamic contrast-enhanced T1-MR imaging. Figure 14 displays the signal-intensity decay of the fibroadenoma detected by those four methods with different *b* values in the eighth slice.

Table 3 shows ADC, signal decay slope, D*, D, and PF parameter values for a fibroadenoma obtained by using different detection methods: K-CEM, fuzzy C-means, K-means, and I-KCEM.

### 4.4. Real Case of Cyst

Breast cysts, which result from fluid accumulation in the breast, have a high incidence rate but are usually non-malignant. A cyst is generally oval or round and feels like a soft grape or a water-filled balloon. Cysts have an internal moisture content of approximately 0.5–0.6 cm.

We consider a patient who was diagnosed with a cyst in the left breast. Figure 15 shows 8th-slice IVIM-DW images and the results of cyst detection by the four methods, as well as the mapping of the cyst region by dynamic contrast-enhanced T1-MR imaging. Figure 16 displays the signal-intensity decay of the cyst detected by the four methods with different *b* values in the 8th slice.

Table 4 shows ADC, signal decay slope, D*, D, and PF parameter values for a cyst obtained by using different detection methods: K-CEM, fuzzy C-means, K-means, and I-KCEM.

## 5. Discussion

### 5.1. Evaluation of the Detection Methods

Dice and Jaccard similarity coefficients were used to evaluate the experimental results. The tumor measurement results were compared to the ground truth, which was manually labeled from the dynamic contrast-enhanced T1-MR images. In addition, we calculated the execution time for tumor detection to further evaluate the efficiency of the four automatic tumor detection methods. The Dice and Jaccard similarity coefficients of each method in mass tumor detection are summarized in Table 5. Table 6 shows the accuracy rate, precision rate, and recall rate for mass tumor detection. Table 7 lists the average execution time of each method for 48 cases in mass tumor detection. The Dice and Jaccard similarity coefficients of each method in non-mass tumor detection are summarized in Table 8. Table 9 shows the accuracy rate, precision rate, and recall rate of non-mass tumor detection. Table 10 lists the execution times of the four methods in non-mass tumor detection.

We found that K-CEM achieved the highest accuracy, according to both the Dice and Jaccard metrics. Table 5 shows that K-CEM can accurately detect and differentiate between different types of breast tumors. In Table 6, we observe a high accuracy rate and precision rate in detecting mass tumors. Table 7 shows the average execution time of each method for 14 mass tumor cases. Table 8 shows the detection of non-mass tumors. Because non-mass tumors are less concentrated tumors, their accuracy of detection will be lower than that of mass tumors. Conversely, in Table 9, the accuracy rate, precision rate, and recall rate are also lower than that for the detection of mass tumors. Table 10 shows the average execution time of each method for nine non-mass tumor cases. The signal strength of the IVIM-MR image decays with different *b* values. Therefore, this hyperspectral signal processing method can be applied to IVIM-MR images.

### 5.2. Quantitative Results in Real Cases

In this study, we tested four types of tumors with a variety of features. All these images were diagnosed by radiologists and physicians. Mass tumors have complete structures and high densities. Non-mass tumors have loose structures and relatively low densities. Very few real cases of fibroadenoma and cyst could be obtained for this study, but these lesions were still included in the table of quantitative results and marked with a star for reference (Table 11). As the diffusion phenomenon of mass is less evident in mass tumors, the ADC, D, and PF values are usually lower in these cases than in the cases of non-mass tumors. Conversely, the diffusion phenomenon in cyst and fibroadenoma cases is usually stronger than in mass and non-mass tumors; therefore, the ADC, D, and PF values are usually larger in these cases than in the cases of mass and non-mass tumors. Figure 17 shows the signal intensity attenuation of different types of tumors using K-CEM. The denser the tumor, the slower the signal intensity decays. The more water content of the tumor, the higher the diffusion coefficient, and the faster the signal intensity decays. For the same type of tumor, all these pixels will have similar parameters. For example, cyst tumor ADC = 2.64 × 10^−3^, Signal Decay Slope = −3.71 × 10^−4^, D* = 4.38 × 10^−3^, D = 1.36 × 10^−3^, PF = 52%, indicating the characteristics of a cyst.

## 6. Conclusions

In clinical diagnosis, DWI and DCE-MRI are the most widely used radiological examinations for the diagnosis of breast lesions. We used hyperspectral target detection method to detect a priori targets with provided known target knowledge (ground truth) as a priori target detection and a posteriori targets with known target signatures (parameters: D, signal attenuation slope, ADC, D*, and PF), but unknown abundance fractions needed to be estimated as a posteriori target detection. In this article, we considered IVIM-DWI with 13 different weighting factors (*b* values) as a 13-band multispectral image cube where each sequence-acquired image is viewed as a spectral band image. Multispectral images were expanded into hyperspectral images using the band expansion process, hence solving the problem of insufficient band images, and improving the classification accuracy. Quantitative analysis was performed by applying five important parameters: D, signal attenuation slope, ADC, D*, and PF. Through these parameters, we can summarize the characteristics of different types of tumors, fibroadenoma, and cyst for each pixel in IVIM-MR images. Even an unexperienced examiner can easily check each pixel from the IVIM image cube and can judge from the quantitative result of the pixel that the pixel is classified as a mass, non-mass, fibroadenoma, cyst, or other. For the same type of tumor, all these pixels will have similar parameters. On the other hand, if an IVIM pixel with similar ADC, D, D*, signal attenuation slope, and PF parameters is detected, it can be said that the pixel is detected as the same type of tumor. We believe that we are the first to use these five parameters to classify tumor tissues. Therefore, the hyperspectral signal processing method can be fully applied to IVIM-MR images. However, the effectiveness of hyperspectral tumor detection methods may possibly be reduced by artifacts and related image alignment problems caused by subject motion and respiratory motion during imaging sequences.

## Figures and Tables

**Figure 1 jpm-11-00656-f001:**
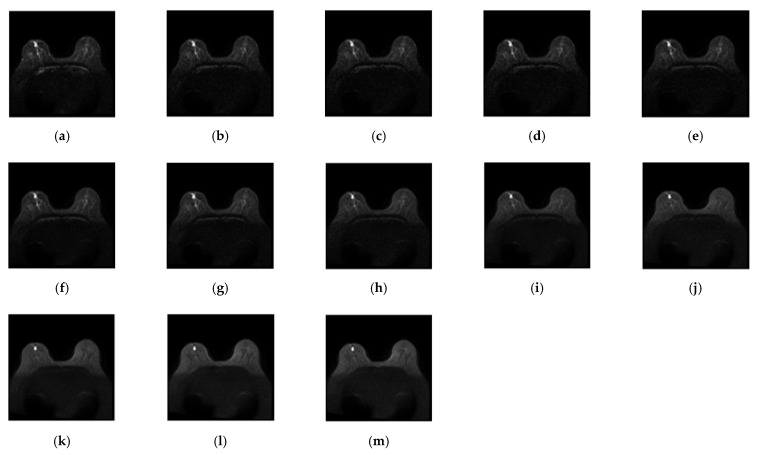
Intravoxel incoherent motion (IVIM) diffusion weight (DW) images with *b* values of (**a**) 0, (**b**) 15, (**c**) 30, (**d**) 45, (**e**) 60, (**f**) 100, (**g**) 200, (**h**) 400, (**i**) 600, (**j**) 1000, (**k**) 1500, (**l**) 2000, and (**m**) 2500 s/mm^2^.

**Figure 2 jpm-11-00656-f002:**
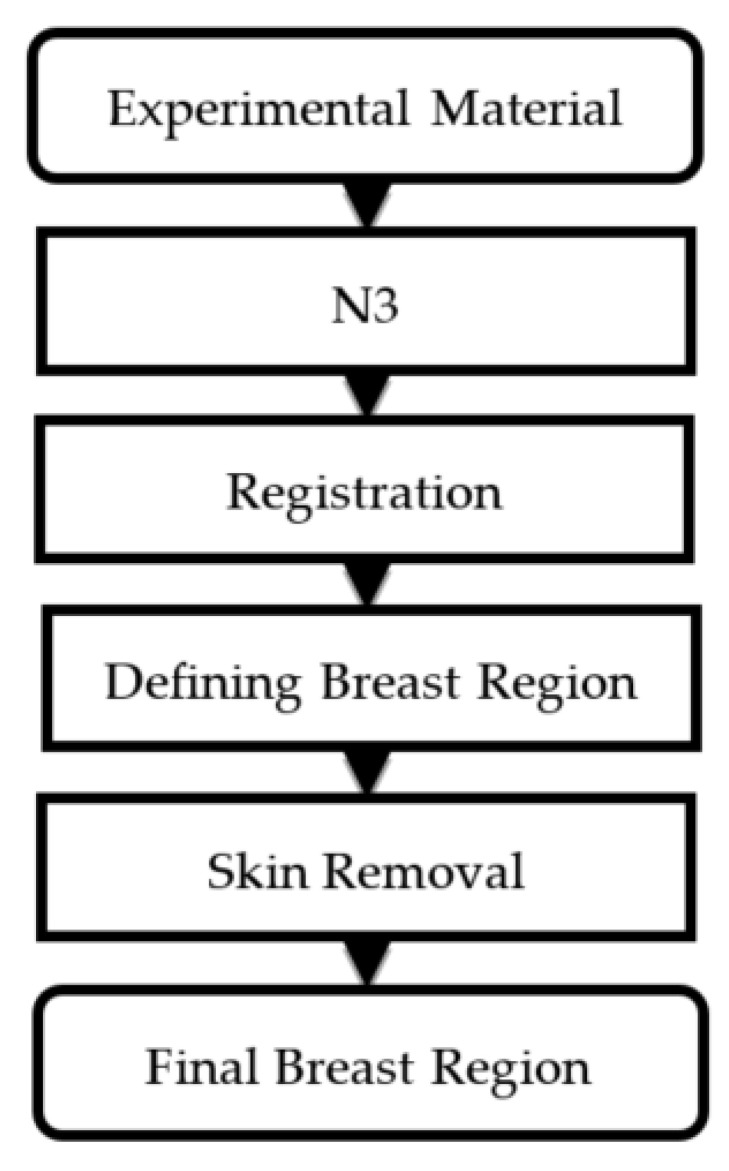
Flow chart showing image pre-processing steps.

**Figure 3 jpm-11-00656-f003:**
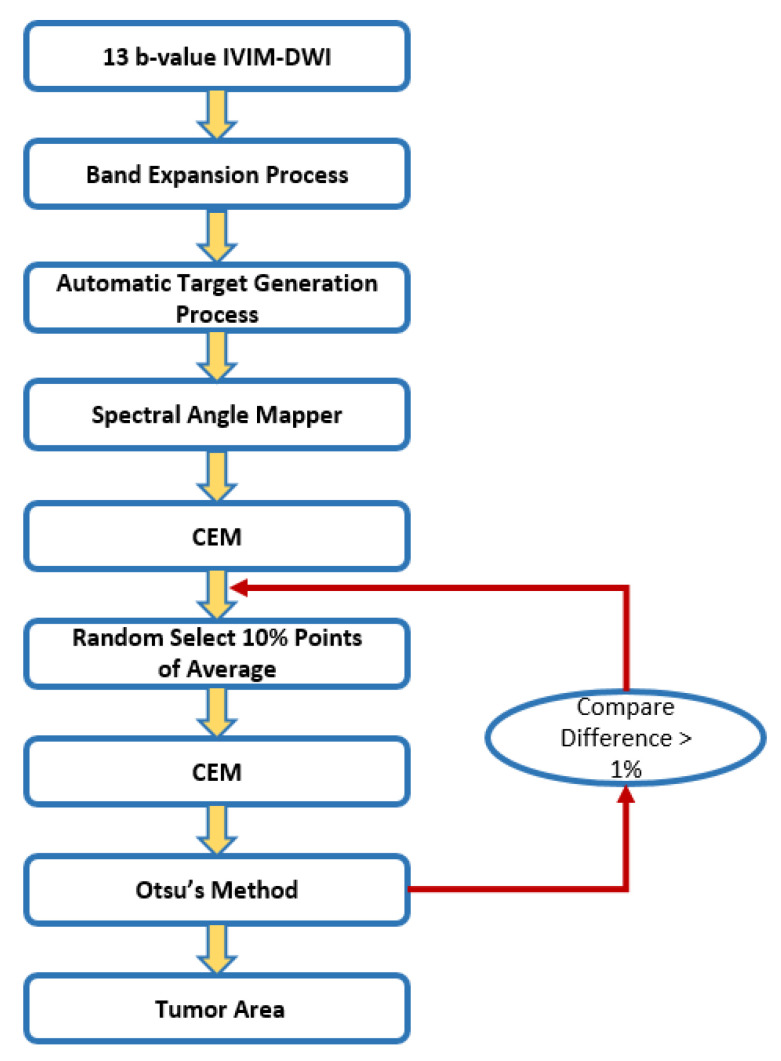
Flowchart of tumor detection via iterative-constrained energy maximization (I-CEM).

**Figure 4 jpm-11-00656-f004:**
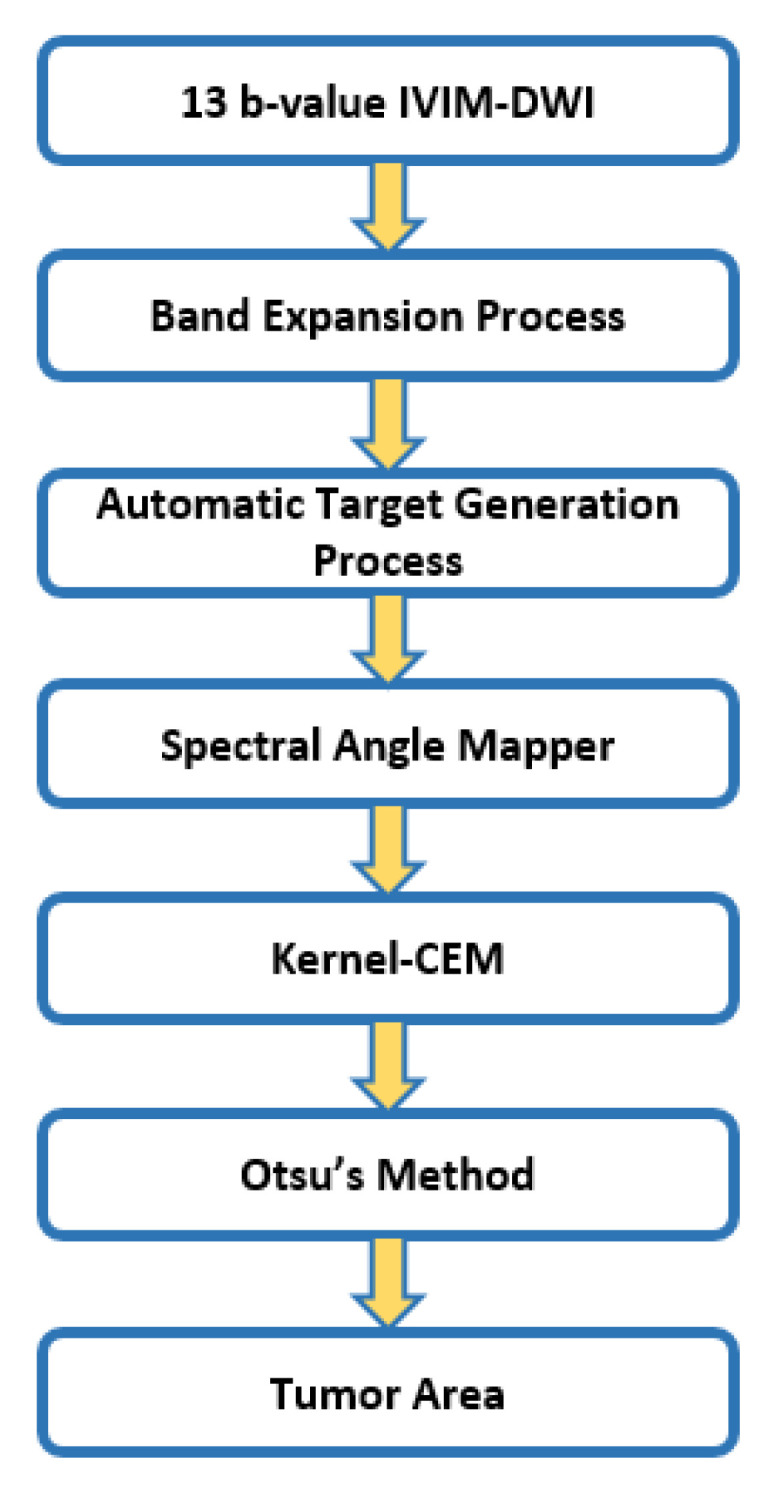
Flowchart of tumor detection via K-CEM.

**Figure 5 jpm-11-00656-f005:**
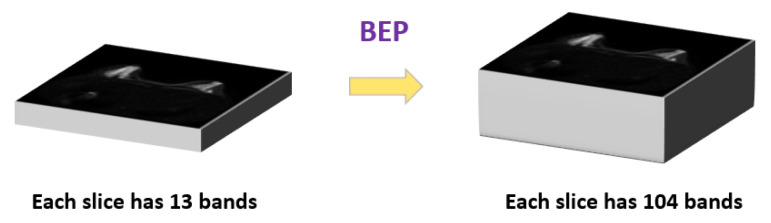
Band expansion process (BEP).

**Figure 6 jpm-11-00656-f006:**
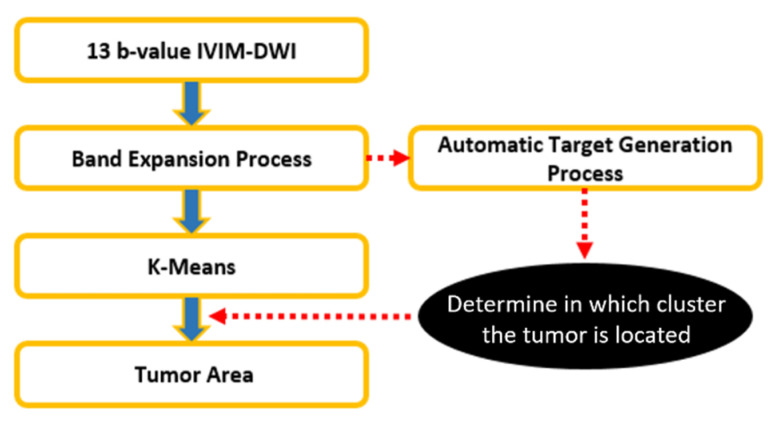
Flowchart of tumor detection via K-means.

**Figure 7 jpm-11-00656-f007:**
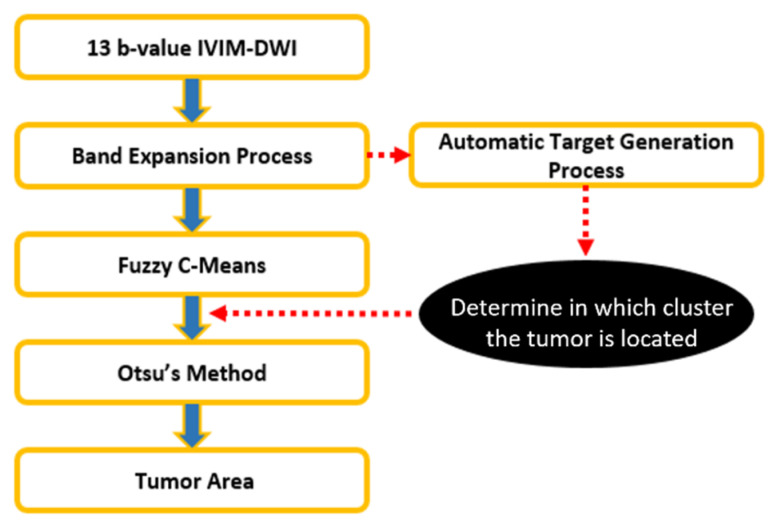
Flowchart of tumor detection via Fuzzy C-means.

**Figure 8 jpm-11-00656-f008:**
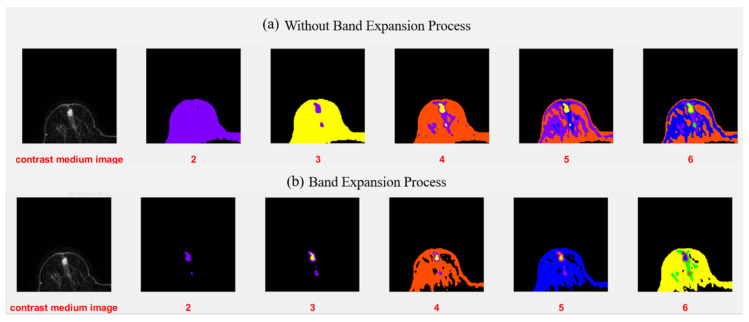
K-means results when *k* = 2–6: (**a**) images without band expansion process; (**b**) images after band expansion process.

**Figure 9 jpm-11-00656-f009:**
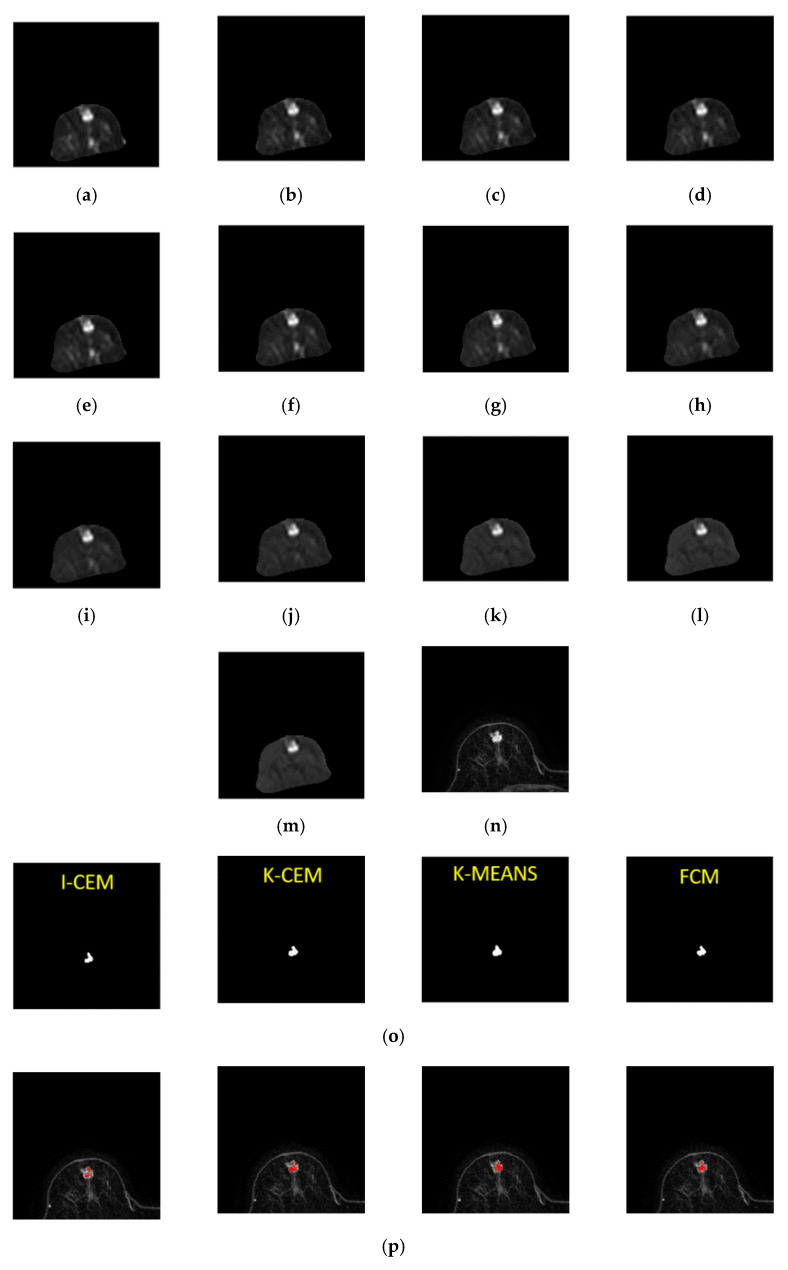
Third-slice IVIM-DW images of a mass tumor with *b* values of (**a**) 0, (**b**) 15, (**c**) 30, (**d**) 45, (**e**) 60, (**f**) 100, (**g**) 200, (**h**) 400, (**i**) 600, (**j**) 1000, (**k**) 1500, (**l**) 2000, and (**m**) 2500 s/mm^2^. (**n**) Dynamic contrast-enhanced T1-MR imaging. (**o**) Tumor detection results obtained via I-CEM, K-CEM, K-means, and FCM. (**p**) Tumors detected via the four methods (left-to-right: I-CEM, K-CEM, K-means, and FCM) mapped onto a dynamic contrast-enhanced T1-MR image.

**Figure 10 jpm-11-00656-f010:**
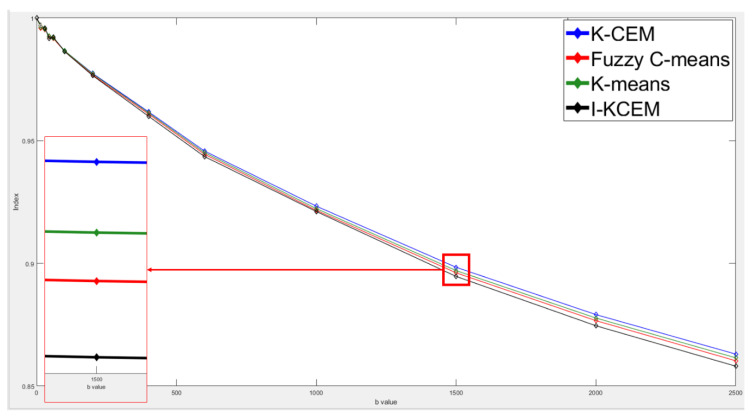
Signal-intensity decay of a mass tumor detected by the four methods using different *b* values in the third slice.

**Figure 11 jpm-11-00656-f011:**
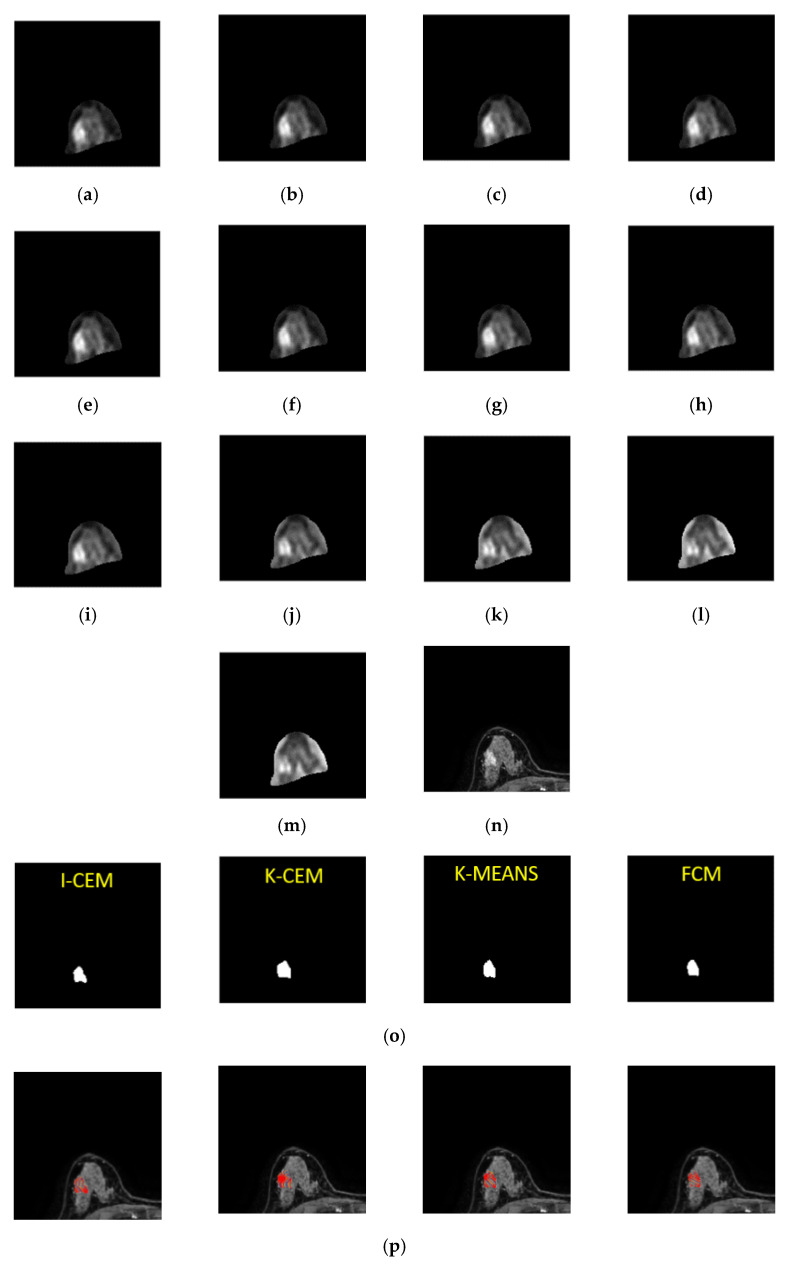
Eighth-slice IVIM-DW images of a non-mass tumor with *b*-values of (**a**) 0, (**b**) 15, (**c**) 30, (**d**) 45, (**e**) 60, (**f**) 100, (**g**) 200, (**h**) 400, (**i**) 600, (**j**) 1000, (**k**) 1500, (**l**) 2000, and (**m**) 2500 s/mm^2^. (**n**) Dynamic contrast-enhanced T1-MR imaging. (**o**) Tumor detection results obtained via I-CEM, K-CEM, K-means, and FCM methods. (**p**) Tumors obtained via the four methods (left-to-right: I-CEM, K-CEM, K-means, and FCM) mapped onto a dynamic contrast-enhanced T1-MR image.

**Figure 12 jpm-11-00656-f012:**
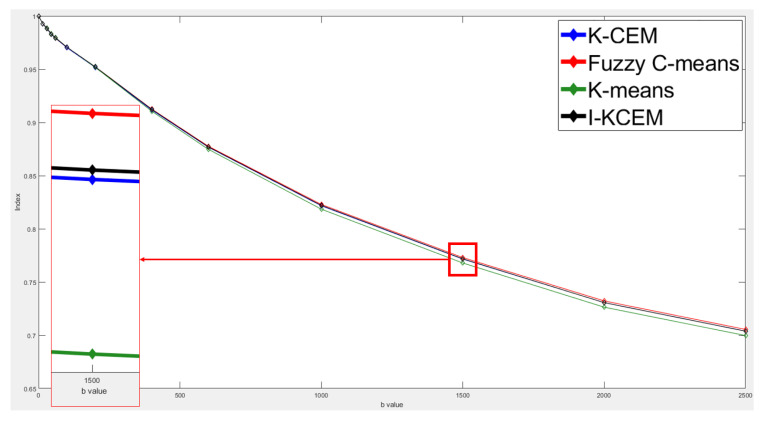
Signal-intensity decay of a non-mass tumor detected by the four methods using different *b* values in the 8th slice.

**Figure 13 jpm-11-00656-f013:**
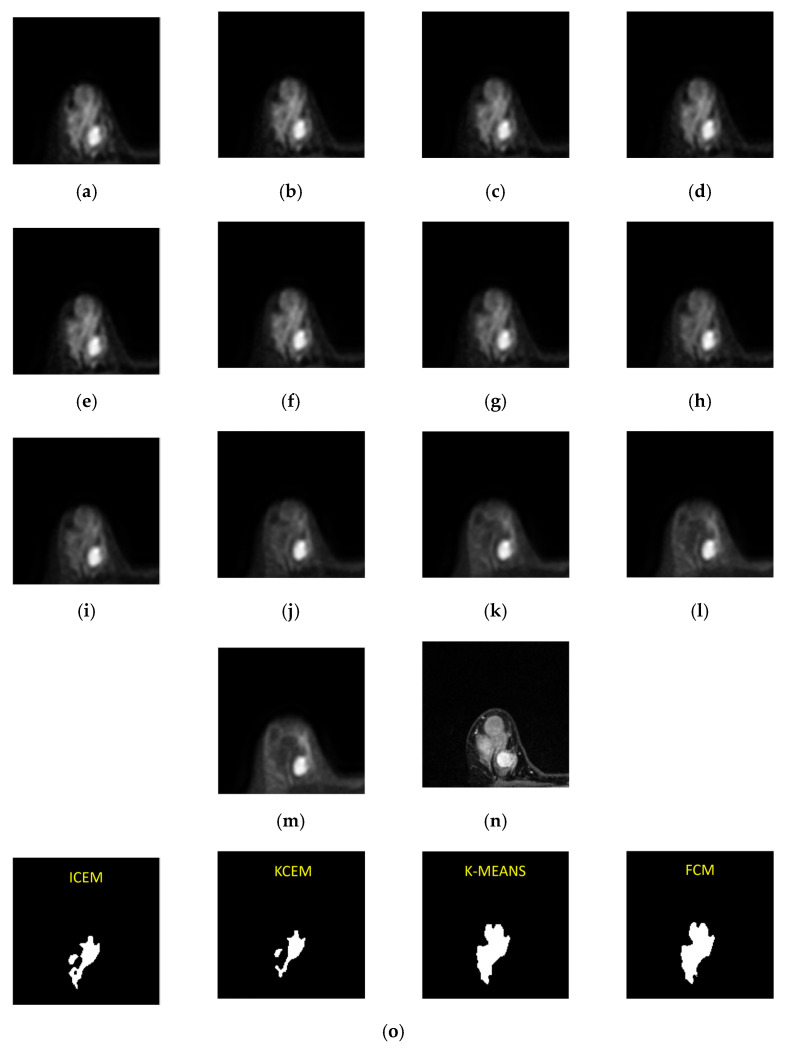
Eighth-slice IVIM-DW images of a fibroadenoma with *b* values of (**a**) 0, (**b**) 15, (**c**) 30, (**d**) 45, (**e**) 60, (**f**) 100, (**g**) 200, (**h**) 400, (**i**) 600, (**j**) 1000, (**k**) 1500, (**l**) 2000, and (**m**) 2500 s/mm^2^. (**n**) Dynamic contrast-enhanced T1-MR image. (**o**) Tumor detection by I-CEM, K-CEM, K-means, and FCM.

**Figure 14 jpm-11-00656-f014:**
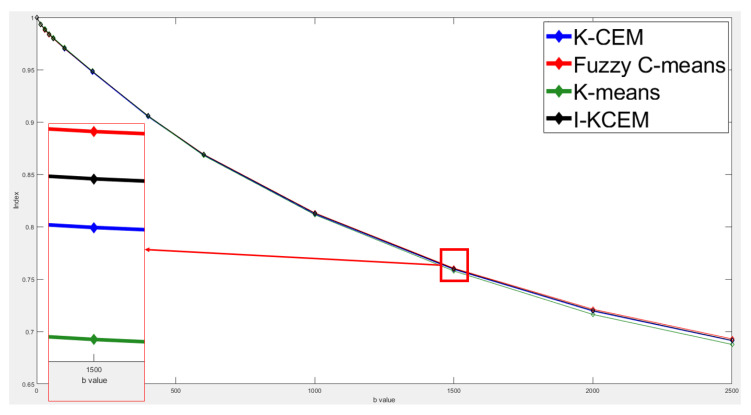
Signal-intensity decay of a fibroadenoma detected by the four methods using different *b* values in the 8th slice.

**Figure 15 jpm-11-00656-f015:**
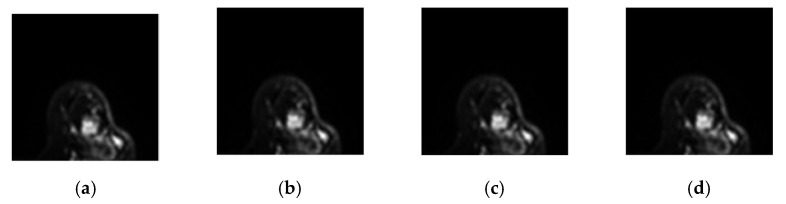
Eighth-slice IVIM-DW images of a cyst with *b* values of (**a**) 0, (**b**) 15, (**c**) 30, (**d**) 45, (**e**) 60, (**f**) 100, (**g**) 200, (**h**) 400, (**i**) 600, (**j**) 1000, (**k**) 1500, (**l**) 2000, and (**m**) 2500 s/mm^2^. (**n**) Dynamic contrast-enhanced T1-MR imaging. (**o**) Tumor detection by I-CEM, K-CEM, K-means, and FCM.

**Figure 16 jpm-11-00656-f016:**
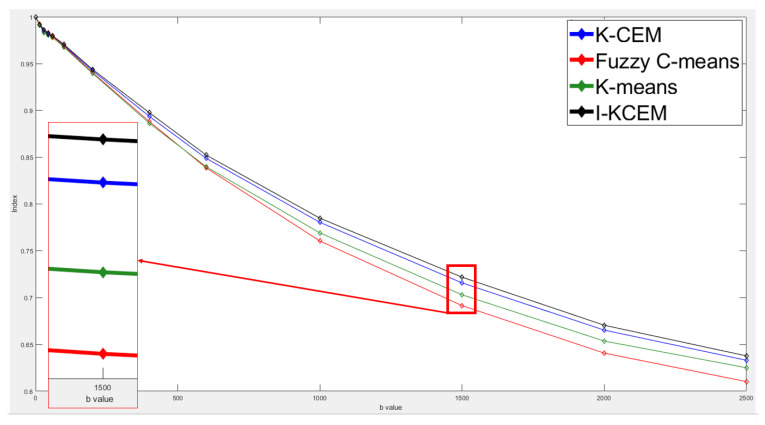
Signal-intensity decay of a cyst detected by the four methods using different *b* values in the 8th slice.

**Figure 17 jpm-11-00656-f017:**
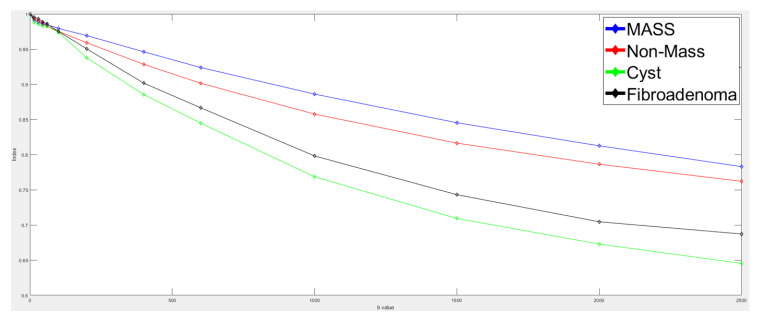
Signal intensity decays using K-CEM.

**Table 1 jpm-11-00656-t001:** Results of applying different detection methods to a mass tumor.

Method Type	ADC	Signal Decay Slope	D*	D	PF
K-CEM	1.20 × 10^−3^	−1.60 × 10^−4^	5.53 × 10^−3^	7.59 × 10^−4^	21%
Fuzzy C-means	1.21 × 10^−3^	−1.68 × 10^−4^	5.86 × 10^−3^	7.43 × 10^−4^	21%
K-means	1.22 × 10^−3^	−1.65 × 10^−4^	6.03 × 10^−3^	7.93 × 10^−4^	22%
I-KCEM	1.21 × 10^−3^	−1.70 × 10^−4^	5.94 × 10^−3^	7.49 × 10^−4^	21%

**Table 2 jpm-11-00656-t002:** Results of applying different detection methods to a non-mass tumor.

Method Type	ADC	Signal Decay Slope	D*	D	PF
K-CEM	1.50 × 10^−3^	−2.16 × 10^−4^	7.58 × 10^−3^	1.07 × 10^−3^	30%
Fuzzy C-means	1.51 × 10^−3^	−2.07 × 10^−4^	7.50 × 10^−3^	0.99 × 10^−3^	32%
K-means	1.51 × 10^−3^	−2.19 × 10^−4^	7.64 × 10^−3^	1.12 × 10^−3^	31%
I-KCEM	1.51 × 10^−3^	−2.10 × 10^−4^	7.85 × 10^−3^	1.19 × 10^−3^	31%

**Table 3 jpm-11-00656-t003:** Results of applying different detection methods to a fibroadenoma tumor.

Method Type	ADC	Signal Decay Slope	D*	D	PF
K-CEM	2.08 × 10^−3^	−3.14 × 10^−4^	4.09 × 10^−3^	1.26 × 10^−3^	45%
Fuzzy C-means	2.10 × 10^−3^	−3.12 × 10^−4^	4.14 × 10^−3^	1.27 × 10^−3^	45%
K-means	2.09 × 10^−3^	−3.15 × 10^−4^	4.44 × 10^−3^	1.26 × 10^−3^	45%
I-KCEM	2.08 × 10^−3^	−3.14 × 10^−4^	4.00 × 10^−3^	1.26 × 10^−3^	46%

**Table 4 jpm-11-00656-t004:** Results of applying different detection methods to a cyst tumor.

Method Type	ADC	Signal Decay Slope	D*	D	PF
K-CEM	2.64 × 10^−3^	−3.71 × 10^−4^	4.38 × 10^−3^	1.36 × 10^−3^	52%
Fuzzy C-means	2.80 × 10^−3^	−3.96 × 10^−4^	4.46 × 10^−3^	1.23 × 10^−3^	49%
K-means	2.78 × 10^−3^	−3.88 × 10^−4^	5.33 × 10^−3^	1.02 × 10^−3^	63%
I-KCEM	2.75 × 10^−3^	−3.63 × 10^−4^	4.56 × 10^−3^	1.19 × 10^−3^	56%

**Table 5 jpm-11-00656-t005:** Mass tumor: Dice similarity coefficient and Jaccard similarity coefficient of each method.

	KCEM	K-Means	Fuzzy C-Means	ICEM
Dice (%)	89.66%	87.22%	87.88%	88.14%
Jaccard (%)	81.25%	77.33%	78.38%	78.79%

**Table 6 jpm-11-00656-t006:** Mass tumor: Confusion matrix.

	KCEM	K-Means	Fuzzy C-Means	ICEM
Accuracy (%)	99.54%	99.35%	99.39%	99.47%
Precision (%)	92.86%	79.45%	80.56%	89.66%
Recall (%)	86.67%	96.67%	96.67%	86.67%

**Table 7 jpm-11-00656-t007:** Mass tumor: Average execution time of each method for 14 cases.

	KCEM	K-Means	Fuzzy C-Means	ICEM
Execution time (s)	22	91.8	1.61	7.48

**Table 8 jpm-11-00656-t008:** Non-mass tumor: Dice similarity coefficient and Jaccard similarity coefficient of each method.

	KCEM	K-Means	Fuzzy C-Means	ICEM
Dice (%)	86.07%	83.50%	83.99%	83.73%
Jaccard (%)	76.09%	73.18%	73.40%	74.02%

**Table 9 jpm-11-00656-t009:** Non-mass tumor: Confusion matrix.

	KCEM	K-Means	Fuzzy C-Means	ICEM
Accuracy (%)	98.41%	97.57%	97.66%	97.74%
Precision (%)	88.42%	82.38%	83.26%	85.16%
Recall (%)	85.42%	87.48%	89.31%	80.58%

**Table 10 jpm-11-00656-t010:** Non-mass tumor: Average execution time of each method for 9 cases.

	KCEM	K-Means	Fuzzy C-Means	ICEM
Execution time (s)	38.385	2.075	3.7	7.594

**Table 11 jpm-11-00656-t011:** Quantitative results for mass, non-mass, fibroadenoma, and cyst tumors.

Tumor Type	ADC	Signal Decay Slope	D*	D	PF
Mass	1.21 × 10^−3^	−1.89 × 10^−4^	6.10 × 10^−3^	0.84 × 10^−3^	23%
Non-mass	1.49 × 10^−3^	−2.14 × 10^−4^	7.53 × 10^−3^	1.03 × 10^−3^	31%
Fibroadenoma	2.08 × 10^−3^	−3.14 × 10^−4^	4.09 × 10^−3^	1.26 × 10^−3^	45%
Cyst	2.64 × 10^−3^	−3.71 × 10^−4^	4.38 × 10^−3^	1.36 × 10^−3^	52%

## Data Availability

The data that support the findings of this study are available on request from the author (S.W. Chan). The data are not publicly available due to restrictions (e.g., their containing information that could compromise the privacy of research participants).

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
