# Peer review of "Quantitative Measurement of Breast Tumors Using Intravoxel Incoherent Motion (IVIM) MR Images"

_jpm, 2021, doi:10.3390/jpm11070656_

Round 1

Reviewer 1 Report

I thank the Authors for submitting a well structured and presented paper on lesion segmentation in breast MRI, using IVIM DWI images.

The main issue with the paper is the lack of a comparison with a reference standard on DWI images (i.e., manual annotations by a radiologist). Using T1 contrast enhanced sequences is useful but not ideal. The portions of a lesion that present contrast enhancement do not have to present restricted diffusion (and the opposite is also true). The comparison between IVIM and DCE can be useful to confirm that the lesion area was correctly identified, but comparing only the 4 segmentation approaches without a "clinical" reference dimishes the usefulness of the reported findings. I would suggest adding this comparison to your experiments.

Also, supervised and unsupervised breast segmentation has been widely investigated in the past, even if your approach indeed presents novel aspects which merit publication. Therefore, some additional discussion/comparison with different approaches could be valuable (e.g., 10.1007/978-3-319-68560-1_63).

Author Response

Reviewer 1 Comments and Suggestions for Authors

  1. I thank the Authors for submitting a well structured and presented paper on lesion segmentation in breast MRI, using IVIM DWI images.

Response:

Thanks for your comments. We will do our best to improve the presentation of this article.

  1. The main issue with the paper is the lack of a comparison with a reference standard on DWI images (i.e., manual annotations by a radiologist).

Response:

Thanks for your comments. All these images were diagnosed by radiologist and physicians. Besides, The ADC was also included in the paper The bi-exponential model of IVIM is defined using the expression.

               Sb / S0  =  (1-PF) exp (- bD)  + PF exp( bD* )

where b is the diffusion sensitization of the MRI sequence; Sb and S0 are signal intensities in pixels with and without the diffusion gradient, respectively; The ADC can be calculated if pf==0 then D=ADC.

The ADC value was measured by at least two DW imaging (DWI) with different signal attenuation at different b values. The values displayed that reflects the degree of diffusion of water molecules through different tissues. According to the following equation:

            ln(Sb/S0) = -b(ADC)   →   ADC=-b EXP(Sb/S0  

  1. Using T1 contrast enhanced sequences is useful but not ideal. The portions of a lesion that present contrast enhancement do not have to present restricted diffusion (and the opposite is also true). The comparison between IVIM and DCE can be useful to confirm that the lesion area was correctly identified, but comparing only the 4 segmentation approaches without a "clinical" reference dimishes the usefulness of the reported findings. I would suggest adding this comparison to your experiments.

Response:

Thanks for your comments. You are absolutely correct. For our experiment, all tumor positions, size ranges, tumor types, etc. of all IVIV-MR medical images were manually marked by experienced radiologists and surgeons with dynamic contrast-enhanced T1-MR. The actual diagnosis of the tumor is also confirmed by ultrasound image examination and by biopsy. The T1 contrast enhancement sequence is only used to draw ground truth maps.

  1. Also, supervised and unsupervised breast segmentation has been widely investigated in the past, even if your approach indeed presents novel aspects which merit publication. Therefore, some additional discussion/comparison with different approaches could be valuable (e.g., 10.1007/978-3-319-68560-1_63).

Response:

Thanks for your comments. The reference paper 10.1007/978-3-319-68560-1_63 proposed an automatic multiple sub-detection for MR breast image segmentation. This paper does a good job in MR breast image segmentation. However, in our paper, we assume that the IVIM-MR breast image segmentation has been completed, and then it will continue our tumor classification process. The main purpose is not to segment, but to classify tumor tissues. Once the tumor is detected, we will check the tumor area pixel by pixel through our detection method. For the same type of tumor, all these pixels will have similar parameters. For example, Cyst tumor ADC=2.64E-03, Signal Decay Slope=-3.71E-04, D*= 4.38E-03, D=1.36 E-03 , PF=52%, indicating the characteristics of a cyst. On the other hand, if an IVIM pixel with similar ADC, D, D*, signal attenuation slope, and PF parameters is detected, then we can say that the pixel is detected as a cystic tumor. We believe that we are the first to use these five parameters to classify tumor tissues.   

Reviewer 2 Report

It is an interesting study. 

I think that the introduction section could be developed and more references added. 

The authors should precise, in the discussions section, the limits of the study in their opinion.  

Also, in the discussions section, the authors should compare the results with those of another MRI techniques and add supplimentar references. 

The conclusions must be concise and strictly resume the results of the study. I suggest authors to rephrase the conclusions. 

Author Response

Reviewer 2 Comments and Suggestions for Authors

  1. It is an interesting study. I think that the introduction section could be developed and more references added.

Response:

Thank you for the opinion, we will rephrase the introduction section and add more references.

  1. The authors should precise, in the discussions section, the limits of the study in their opinion. Also, in the discussions section, the authors should compare the results with those of another MRI techniques and add supplementary references. The conclusions must be concise and strictly resume the results of the study. I suggest authors to rephrase the conclusions.

Response:

Thanks for your comments. We reformulated the discussion part and the conclusion part, and tried our best to improve the presentation of this article.

Here is the conclusion:

  1. Conclusions

In clinical diagnosis, DWI and DCE-MRI are the most widely used radiological examinations for the diagnosis of breast lesions. We use hyperspectral target detection method to detect, a priori targets with provided known target knowledge (ground truth) as a priori target detection and a posteriori targets with known target signatures (parameters: D, signal attenuation slope, ADC, D* and PF), but unknown abundance fractions needed to be estimated as a posteriori target detection. In this article, we considered IVIM-DWI with 13 different weighting factors (b values) as a 13-band multispectral image cube where each sequence-acquired image is viewed as a spectral band image. Multispectral images were expanded into hyperspectral images using the band expansion process, hence solving the problem of insufficient band images, and improving the classification accuracy. Quantitative analysis is performed by applying five important parameters: D, signal attenuation slope, ADC, D* and PF. Through these parameters, we can summarize the characteristics of different types of tumors, Fibroadenoma, and Cyst for each pixel in IVIM-MR images. Even an unexperienced examiner can easily check each pixel from the IVIM image cube, and can judge from the quantitative result of the pixel that the pixel is classified as a mass, non-mass, fibroadenoma, cyst, or others. For the same type of tumor, all these pixels will have similar parameters. On the other hand, if an IVIM pixel with similar ADC, D, D*, signal attenuation slope, and PF parameters is detected, it can be said that the pixel is detected as the same type of tumor. We believe that we are the first one to use these five parameters to classify tumor tissues. Therefore, the hyperspectral signal processing method can be fully applied to IVIM-MR images. However, the effectiveness of hyperspectral tumor detection methods may possibly be reduced by artifacts and related image alignment problems caused by subject motion and respiratory motion during imaging sequences.

Round 2

Reviewer 1 Report

I thank the Authors for addressing my previous comments. The technical novelty of the paper has been better highlighted.